# Disordered Residues and Patterns in the Protein Data Bank

**DOI:** 10.3390/molecules25071522

**Published:** 2020-03-27

**Authors:** Mikhail Yu. Lobanov, Ilya V. Likhachev, Oxana V. Galzitskaya

**Affiliations:** 1Institute of Protein Research, Russian Academy of Sciences, Pushchino, 142290 Moscow, Russia; m.u.lobanov@mail.ru (M.Y.L.); ilya_lihachev@mail.ru (I.V.L.); 2Institute of Mathematical Problems of Biology, Keldysh Institute of Applied Mathematics, Russian Academy of Sciences, Vitkevicha str.1, Pushchino, 142290 Moscow, Russia; 3Institute of Theoretical and Experimental Biophysics, Russian Academy of Sciences, Pushchino, 142290 Moscow, Russia

**Keywords:** disordered residues, identity, homo-repeats, protein structure, low complexity regions

## Abstract

We created a new library of disordered patterns and disordered residues in the Protein Data Bank (PDB). To obtain such datasets, we clustered the PDB and obtained the groups of chains with different identities and marked disordered residues. We elaborated a new procedure for finding disordered patterns and created a new version of the library. This library includes three sets of patterns: unique patterns, patterns consisting of two kinds of amino acids, and homo-repeats. Using this database, the user can: (1) find homologues in the entire Protein Data Bank; (2) perform a statistical analysis of disordered residues in protein structures; (3) search for disordered patterns and homo-repeats; (4) search for disordered regions in different chains of the same protein; (5) download clusters of protein chains with different identity from our database and library of disordered patterns; and (6) observe 3D structure interactively using MView. A new library of disordered patterns will help improve the accuracy of predictions for residues that will be structured or unstructured in a given region.

## 1. Introduction

Intrinsically disordered proteins and regions are very important for many eukaryotic cell processes [1,2,3,4,5,6]. Recently, interest in intrinsically disordered regions has only increased since RNA-binding proteins with prion-like domains such as FUS, TDP-43, and others with large intrinsic disordered domains are involved in processes such as liquid-gel phase transition [7,8,9,10]. Virus shell proteins also have disordered regions that may be important for antiviral vaccine development [11,12,13]. According to the authors of the paper, the most disordered group (A) included coronaviruses that had the highest respiratory transmission [14].

It has been demonstrated that the functions of intrinsically disordered regions are both length- and position-dependent [15]. However, the functional importance of many disordered regions and patterns remains unclear. In order to obtain the correct statistics of disordered residues and create a library of disordered residues and patterns, we should perform Protein Data Bank (PDB) clustering, which is a necessary procedure for processing big data.

There are different tools for clustering. Some of them only have APIs (application program interfaces) for programmers or files for uploading to FTP servers such as MODELLER [16], Prody [17], and MaxCluster (http://www.sbg.bio.ic.ac.uk/maxcluster). The main purpose of the MaxCluster software is clustering and alignment by using the geometry information, not only sequences. In our case, we had a web-interface as well as complete files in the text and SQL-dump formats. Some programs had their own cluster division. The YAKUSA project [18] only has three clusters for PDB: 90%, 70%, and 50%. The PDBFlex [19] database explores the intrinsic flexibility of protein structures by analyzing structural variations between different depositions and chains in asymmetric units of the same protein in the PDB. This allows for regions and types of structural flexibility present in a protein of interest to be easily identified. Structures of protein chains with identical sequences (sequence identity > 95%) were aligned, superimposed, and clustered. We grouped all protein chains in the PDB into five clusters nested in each other: 100%, 75%, 50%, 25%, and 5%. We believe that such separation is sufficient for a wide variety of tasks and studies.

Moreover, there are only a few software solutions that work with a whole protein data bank: MobiDB [5], YAKUSA [18], and ProDy [17]. It is necessary to create the clustered PDB, which is important for analyzing disordered residues and building a library of disordered patterns. In addition, the clustered PDB simplifies the process of filtering protein structures during their analysis and searches for common structural characteristics among non-identical proteins.

The first library of disordered motifs was published in 2010 [20]. In this work, we present the fifth version of this library, built according to the new rules. Only the first and second versions have been published [20,21]. The third version was obtained during the development of the IsUnstruct program [22], and the fourth after a long break in 2018, but which was not published. The fifth version of the library obtained in this paper with the new rules for constructing candidates, taking into account the homo-repeats and patterns consisting of two types of amino acid residues. The two types of patterns were noted in the paper: the proline-rich patterns and the charged patterns [23].

We have created the largest library of disordered patterns combining motif discovery and identification of a disorder protein segment in the clustered PDB. We used the clusters of protein chains, where the identity between the chains within the cluster exceeds 75%. Figure 1 presents data for the PDB clustering in the different years to show how the quantitative composition of the main clusters has changed. The new version of the library includes 518 unique disordered patterns and 1214 patterns consisting of two kinds of amino acids.

## 2. Results and Discussion

### 2.1. Search for Disordered Patterns Consisting of Two Types of Amino Acid Residues

These patterns, like homo-repeats, are sequences with a low level of complexity. They consist of any pair of amino acids (for example: GSGSGSGSGSSG, GAGAGGAGAG, APAAAPPA, DDDDEDDDE).

It is easy to evaluate how often such sequences should occur in random sequences. Take a random sequence of a length L. The probability that it will consist of only two amino acids is $P_{AC2}\left(L\right)=\overset{20}{\underset{i=1}{\sum }}\overset{20}{\underset{j=i+1}{\sum }}\left({\left(p_{i}+p_{j}\right)}^{L}-p_{i}^{L}-p_{j}^{L}\right)$. Here, $p_{i}$,$p_{j}$ are the amino acid frequencies. In the simplest case, we assume that all amino acids occur with the same frequency of 0.05. Then, the equation is simplified to the form: ${\tilde{P}}_{AC2}\left(L\right)=190*\left(0.1^{L}-2*0.05^{L}\right)$. For length 8, we get $P_{AC2}\left(8\right)=9*10^{-6}$ and ${\tilde{P}}_{AC2}\left(8\right)=2*10^{-6}$. The frequencies were calculated from the 122 proteomes.

The statistics were collected on the clustered PDB, on the set of 122 proteomes, and tested on the random proteome (see Methods). Everywhere, we counted how many times the patterns were encountered, and in the PDB, we looked at the fraction of the disordered residues. It turned out that the patterns of length 7 residues and longer are mostly unfolded (N_u_–N_f_ > 0), shorter ones are mostly ordered (see Table 1 and Figure 2). As can be seen from Figure 2, the optimal length of the patterns is 8. The patterns of this length formed the basis of our database.

We analyzed the occurrence of the AC2 patterns in the 122 (97 eukaryotic and 25 bacterial) proteomes and the random proteome. Interestingly, in the random proteome, there were about 40 thousand residues (40,521) in the AC2 patterns, while in the real proteomes, there were more than four million (4,447,128). That is, in the real proteins, AC2 sequences were 100 times more than in the random ones. If we multiply the previously calculated P_AC2(8)_ by the number of residues in the proteome base, we get 48 thousand residues. This is in a good agreement with the data for the random proteome. Of particular interest are the AC2 sequences with internal repetition: dipeptide repeats (01) and four-peptide repeats (0001, 0010, 0011, 0100, 0110, 0111). Symbols 0 and 1 conceal any pair of amino acids. In the proteomes, such patterns covered 585,638 residues, and in the random sequences 2569, that is, the AC2 sequences with an internal repeat occurred 300 times more often than expected. In the real proteins from the proteomes, every fourth protein contains the AC2 sequences. Or rather, 331,279 real proteins out of the 1,449,561 present in the database contained the AC2 sequences.

Frequency analysis can also be applied to the clustered PDB. If sequences of length of 8 or more were to occur in the PDB by chance, we would expect to find about 100 residues in such sequences. In reality, we saw 8861 residues, or a couple of orders of magnitude greater than expected.

### 2.2. Occurrence of Patterns Consisting of Two Amino Acids in the Clustered PDB

As above, we considered all sequences consisting of only two types of amino acids with the length of8 or more. It is easy to see that the most common pair was GlySer. Table 2 shows that sequences consisting only of a given pair of amino acids are usually disordered and contain one third of the sequences considered. Such sequences typically contain glycosylation and phosphorylation sites (http://elm.eu.org/) [24]. It is interesting to note that sequences with GlySer were found both at the N- and C-ends of the protein chain, and in the middle without much preference. The bright example of GlySer patterns was the FUS protein [10,25]. For AlaGly and AlaPro, this site did not find anything.

### 2.3. DisResClustered DataBase

The DisResClustered database is available at http://oka.protres.ru/cluster_pdb/. This database contains only the x-ray structures. In the database, the protein chains are combined in the CLUSTERS with different levels of identity: 5%, 25%, 50%, 75%, and 100%. The user can see the number of entities in the 5%, 25%, 50%, 75%, and 100% clusters. To find the PDB structure at the **HOME tab**, the user should specify the corresponding PDB entry (in the standard 4-symbol format, for example, PDB entry 1edh) and press Search or Enter. In this case, cadherin (1edh chain A) has 79 sequences with 5% identity, 48 sequences with 25% identity, 27 sequences with 50% identity, 22 sequences with 75% identity, and only two identical sequences with 100% identity. The user will be redirected to the **Selected Group tab** to see the 5%, 25%, 50%, 75%, and 100% identity groups by pressing the mouse in the corresponding column. If they choose the 75% column, the user will see 22 chains. To find disordered regions in the selected protein chain, the user will be redirected to the **Disordered Residues tab**. Undefined amino acid residues are highlighted by a yellow color and are labeled as U (unstructured). To look at the PDB structure, the user will be redirected to the **3D-View tab**. If the user is interested in some disordered patterns or motifs (homo-repeats, etc.) in the clustered PDB, it is possible to use the function of searching. The flowchart presented in Figure 3 illustrates the operation of the database.

### 2.4. Implementation: Disordered Patterns from the Clustered PDB

Considering the 75% level of identity, we created a new version of the library of disordered patterns. At present, the library includes 518 disordered unique patterns (version of 16 January 2019) in comparison with 141 patterns obtained in 2010 (version of 28 June 2010) [21] (Table 3). The patterns occur more often as short fragments. Patterns of five–six–seven residues long occurred more frequently (416 out of 518) among the disordered patterns of the library. Moreover, the library includes 1214 disordered patterns consisting of two kinds of amino acids. The user can download the list of disordered patterns. At the level of 75% identity, we performed the statistics of disordered residues for all 20 amino acid residues in the N- and C-termini and in the middle part of the protein chains (see Figure 4 and figures in the Statistics tab).

### 2.5. Occurrence of Disordered Patterns in the Proteomes

Our database has 518 unique patterns. Of these, 373 contain five non-X residues and 83 contain six non-X residues, and there are even five patterns containing four non-X residues. It is not difficult to calculate the expected number of patterns in the proteomes: $N_{o}\approx 6.3*10^{8}*{\left(1/20\right)}^{L}\;$. Here, $6.3*10^{8}$ is the number of residues in the proteomes and *L* is the effective length (without X) of patterns. For *L* = 5, we expect the pattern to occur in the proteomes about 200 times, for *L* = 6, about 10 times. For each template, we calculated $N_{o}$, taking into account the real frequency of amino acids in the proteomes as well as $N_{p}\;$ as the number of proteins with the given pattern. A total of 298 patterns were found at about the same frequency as we expected, and 220 were more often than expected (data on the site).

It was interesting to observe and compare the independence of the pattern from the histidine tag in the PDB and the frequency of occurrence of the pattern in the proteomes. Among others, we had the patterns GxxxHHHH (*N_p_*:*N_o_* = 1248:14), HHHHxxxS (1821:18), AxxxHHHHHH (678:0.4), HHHHHxxxP (904:0.3), and others directly related to the histidine tag. As can be seen, such patterns occurred in the proteomes significantly more often than expected. This is due to the fact that homo-repeats are more common in proteomes than would be expected from the frequency of amino acids (Lobanov et al., 2016). Moreover, the homo-repeats of histidine are also quite common. Here, it is important only to emphasize that in the PDB, almost all histidine homo-repeats were the artificial additives, and were natural motifs in the proteomes. Patterns associated with the histidine tag in the PDB, but not containing parts of the histidine tag, were found in the proteomes at approximately the same frequency as expected: ENLxFQ (178:233), ASxTxxxxMGR (22:20), and LVPRGS (56:66).

It is also interesting to consider other patterns that were more common than expected: DDDDK (1346: 296), GxSGSSG (837: 98), SPxxSPT (4018: 726), GxxGxxGGGxG (9770: 52), EEEED (14402: 559), APAxxxAP (6913: 1059), PxPAxxPA (6971: 721), and others. It was easy to catch the pattern associated with these patterns. As can be seen, they consist of a pair of residues and non-comparable items (X). In other words, we again caught the pattern associated with sequences of low complexity.

## 3. Materials and Methods

### 3.1. Construction of Clustered Protein Data Bank

We examined all protein structures determined by x-ray diffraction analysis with a resolution higher than 3 Å and a protein size greater than or equal to 40 amino acid residues, published in the PDB (version dated January 16, 2019); 150,912 PDB entries contained 277,583 protein chains. In the first stage, these 277,583 chains could be divided into 74,378 classes. We called these classes clusters with 100% identity (C100). In the second stage, we created the clusters of chains with the identity within each cluster of ≥75% (C75). Identity was calculated by the equation: *Id* = *I*/(*L*_1_ + *L*_2_ − *I*) × 100%, where *I* is the number of identical residues and *L_1_* and *L_2_* are the numbers of amino acid residues in each considered protein. To calculate the identity, we used the BLAST server with the default parameters [26]. Then, the C75 clusters were combined into the clusters with the identity of *Id* ≥ 50%, etc. The dependence of the number of clusters on the identity between the chains inside the cluster is presented in Figure 1.

### 3.2. Statistical Analysis of Disordered Residues

The statistical analysis of the disordered residues was performed taking into account 74,378 unique protein chains taken from the PDB database. In this database, 4.75% of the residues were disordered in the x-ray structures. For statistics, we used the clusters of protein chains, where the identity between the chains within the cluster exceeded 75% (37,205). There were 10,149,440.5 amino acid residues in the C75 clusters, if we summarize the average chain lengths in each cluster. That is, the average protein chain length consisted of 273 amino acid residues. Statistics for disordered residues were obtained separately for the N- and C-ends as well as for the central part of the protein chains. It was shown that the frequencies of occurrence of the disordered residues of 20 types at the ends of the protein chains differed from those in the middle part of the protein chain.

For statistical analysis, we examined all chains, but with different weights. We considered averaged data for clusters with Id ≥ 75% (C75). To do this, we first averaged within clusters with Id = 100% (C100), and then averaged within the C75 cluster. For example, the SSPAK pattern (SerSerProAlaLys) identified disordered residues in the five C75 clusters. Let us consider one of them. Cluster C75 contains two C100 sub-clusters: the first includes 1hynQ, 1hynR, 1hynP, and 1hynS; the second 4ky9P, 4ky9A. In each of the sequences, the unfolded and folded residues (U, S) are covered by the patterns: (23, 8), (24, 7), (31, 0), (31, 0) in the first one, and (8, 0), (6, 2) in the second. Averages inside the C100 are U = 27.25 and S = 3.75 in the first, and U = 7, S = 1 in the second sub-cluster. As a result, in this cluster, the SSPAK pattern allocated an average of 17.13 disordered and 2.37 ordered residues, (http://bioproteom.protres.ru/cluster_pdb/downloads/SSPAK.html).

### 3.3. Construction A Library of Disordered Patterns

#### 3.3.1. Patterns Consisting of Two Types of Amino Acids

We examined all variants of patterns consisting of two types of amino acids (AC2). Among all of these patterns, three groups could be distinguished: homo-repeats with an adjacent single amino acid (011111111, 10000000); with internal repeats (01010101, 0001001, 00110011, etc.); and all the rest. Symbols 0 and 1 concealed any pair of amino acids. In total, 1214 patterns were found.

#### 3.3.2. Unique Disordered Patterns

To create a library of disordered patterns, we considered the two-stage procedure using the clustered PDB by applying simpler rules than in our previous work [20,21]. In the first stage, we created a list of candidate patterns. To be a candidate for a pattern, the considered pattern should be disordered in half the cases among chains from a cluster with 100% identity. We did not limit the length of the patterns. When we found the candidate in the patterns, we checked whether this candidate had met us before. If the pattern had been encountered, then it was discarded. Next, we searched for the homologues of each candidate in patterns with an identity above 80% for the entire database. That is, if the length of the candidate was 3–5 residues, then we did not look for homologues. If the length was 6–10 residues, then one substitution was allowed in the homologue, two substitutions for a length of 11–15 residues, and so on. When the homologue was found, we designed a new pattern with the ability to select any residue in positions diverging between the candidate and its homologue. We marked the discrepancies with the letter X. Again, we checked if we had selected the generated template previously. If not, then the generated template was added to the list of candidates. If we found several homologues diverging in different positions, then the combined template was added to the list of candidates in the patterns. We will explain all the above by using an example. Therefore, we found the following candidates and homologues:candidate   ASMTGGQQMGR (3lpjA)homolog   ASMTGGgQMGR (4wzhB)homolog   ASMTGGnnMGR (3r0rA)homolog   ASMTssQQMGR (5aq5B)homolog   ASnTGGQQMGR (4nv7B)

We formed a list of candidates with replacement options.

candidate   ASMTGGQQMGR (3lpjA)homolog   ASMTGGnnMGR (3r0rA)X-candidate   ASMTGG **XX** MGR candidate   ASMTGGQQMGR (3lpjA)homolog   ASMTssQQMGR (5aq5B)X-candidate   ASMT **XX** QQMGR candidate   ASMTGGQQMGR (3lpjA)homolog   ASnTGGQQMGR (4nv7B)X-candidate   AS **X** TGGQQMGR

Then, we created the list of candidates for patterns by combining the X-patterns.

X-candidate  ASMTGG **XX** MGRX-candidate  ASMT **XX** QQMGRX-candidate  ASMT **XXXX** MGR synthesis X-candidate  ASMT **XXXX** MGRX-candidate  AS **X** TGGQQMGRX-candidate  AS **X**T **XXXX** MGR synthesis

As a result, we received six patterns for further work:ASMTGGQQMGRASMTGG**XX**MGRASMT**XX**QQMGRAS**X**TGGQQMGRASMT**XXXXX**MGRAS**X**T**XXXX**MGR


The last option was selected by us. For simplicity, we skipped several options for combining. Then, 2,746,944 candidates in the disordered patterns were selected for the candidate list.

We can say that pattern *A* matches the chain *P* at the position *s*, if all residues of the pattern coincide with the residues of the sequence, except for the residues marked with the letter X. Next, we considered the following terminology: N_u_ is the sum of the average number of unfolded residues in clusters with 75% identity (C75); N_f_ is the sum of the average number of ordered residues in the clusters with 75% identity matched by the pattern; C_u_ is the number of clusters with an identity of 75% in which N_u_ > N_f_; C_f_ is the number of clusters with an identity of 75% in which N_u_ ≤ N_f_. It should be noted that N_u_ and N_f_ (in the pattern) corresponded to the number of disordered and ordered residues if each 75% cluster had one sequence. Protein *P* has an occurrence of pattern *A*, if *A* matches *P* at position s. Fragment *A* = *P_j_* [s + 1, s + L] of chain *P_j_* is considered as a candidate disordered pattern if it meets the following conditions: (C1) C_u_ ≥ 5; (C2) C_u_ > C_f_; (C3) N_u_ > N_f_.

There were 198,382 patterns satisfying conditions C1, C2, and C3. After choosing homo-repeats and AC2 patterns, we had 177,964 patterns left. In the next step, we selected disordered patterns from the candidate list using the following iterative greedy procedure. Of the 177,964 patterns, we selected a pattern with a maximum value of D = N_u_ − N_f_. Then, for the remaining patterns, the values of N_u_, N_f_, C_u_, and C_f_ were recalculated without taking into account the residues matched by the first pattern. Again, all other patterns were checked for compliance with conditions C1, C2, and C3. Among the remaining patterns satisfying conditions C1, C2, and C3, a pattern with a maximum value of D was selected. If there were no patterns satisfying conditions C1, C2, and C3, the procedure was stopped. The iterative procedure was stopped when 1329 patterns were selected (D > 0). Finally, we were interested in patterns for which D ≥ 25 (a value of 25 corresponds to summing the weights of five whole disordered patterns with five residues in length in five clusters without neighboring regions, or terminal occurrence). There were 518 of such patterns.

It is no secret that very often, the residues necessary for protein purification, the so-called histidine tag (H6), are inserted in the proteins. Other patterns are also often added along with the histidine tag. Since the additives are artificial, they, as a rule, are not ordered and are not visible during x-ray analysis. Therefore, our method inevitably includes such regions in our library of disordered residues. To separate natural patterns from artificial additives, we studied how often there were less than 40 amino acid residues between the found patterns and histidine tag. We calculated how many percent (PC) cases the pattern occurred near the histidine tag (Table 4).

We cannot argue that all the patterns linked to the histidine tag are really artificial additives. Some of the patterns can be found simply at the *N*- or *C*-terminus of the protein, and it is there that researchers add the histidine tag for routine protein purification. However, it is logical to assume that they are still usually added artificially.

### 3.4. Real and Random Proteomes

Statistics for patterns consisting of two types of amino acids were collected on the 122 proteomes [27], and the random proteome. In the random proteome, the lengths of the sequences completely coincided with the lengths of the sequences in the real proteomes. The frequencies of the amino acids were taken from the proteomes. However, the sequences themselves are random (1,449,561 random sequences).

## 4. Conclusions

Now, we understand that cells use their own intrinsic disordered proteins and disordered regions for communication and many functions. It has been demonstrated that the occurrence of simple motifs can be the imprints of evolutionary history [28,29,30]. It was necessary to create a new library of disordered patterns using the clustered PDB, which is important for analyzing disordered residues in homologous proteins with different degrees of identity.

We carried out PDB clustering, which now allows us to work with one of the representatives of C100, C75, C50, C25, C5, or to take all the chains, but to do averaging inside the clusters.

For the first time, we created the patterns consisting of two types of amino acids in a separate group. The AC2 patterns were much more common than expected for random sequences. Starting with a length of 8 residues, such patterns were predominantly disordered. The most common patterns in the PDB were made up of a GlySer pair. The new version of the library of disordered residues and disordered patterns has been built. The data are summarized in one database (Figure 5) http://bioproteom.protres.ru/cluster_pdb/.

It can be assumed that the found patterns may have different biological functions, the search for which will be dealt with in the future.

## Figures and Tables

**Figure 1 molecules-25-01522-f001:**
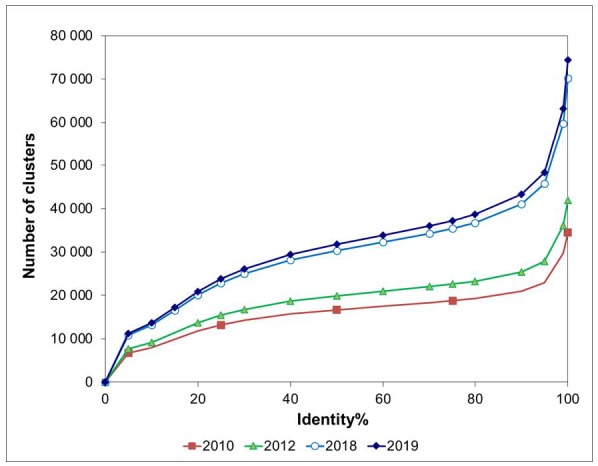
Dependence of the number of clusters on the identity between protein chains for the different years of Protein Data Bank (PDB) clusterization.

**Figure 2 molecules-25-01522-f002:**
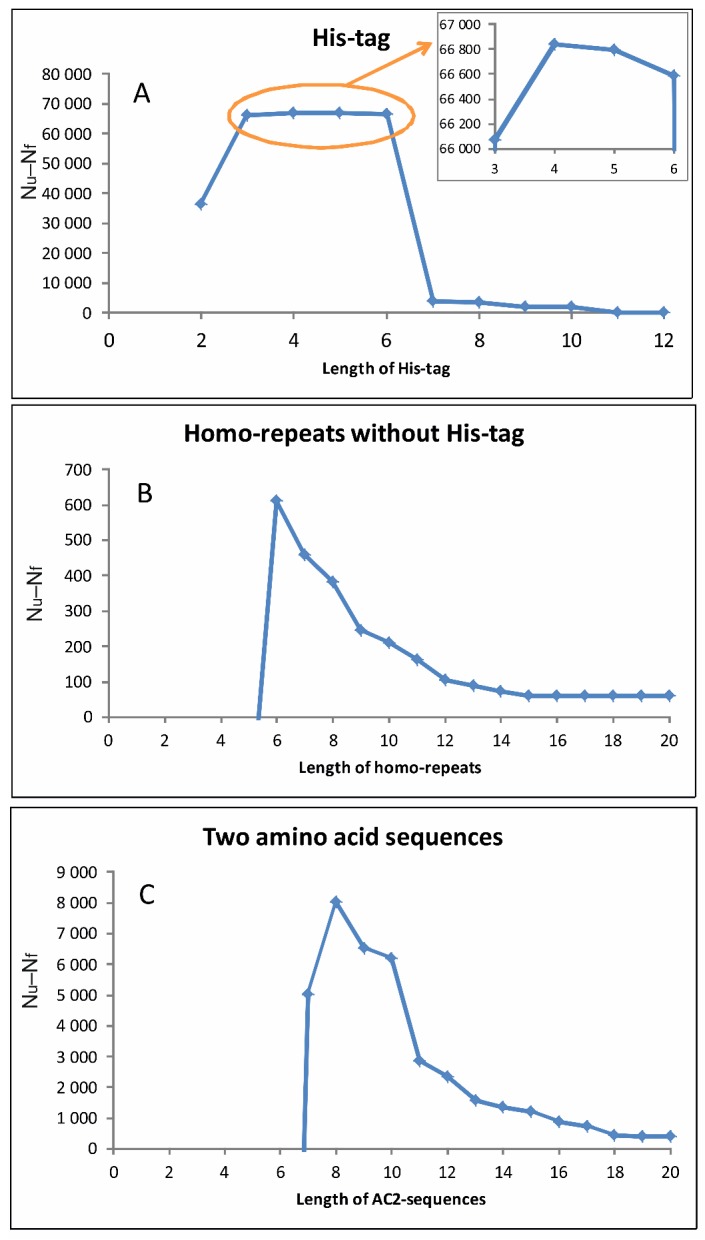
Dependence of the difference in the number of disordered and ordered residues in sequences covered by the patterns on the length of the last: (**A**) histidine tags, (**B**) homo-repeats, and (**C**) AC2-sequences in the clustered PDB.

**Figure 3 molecules-25-01522-f003:**
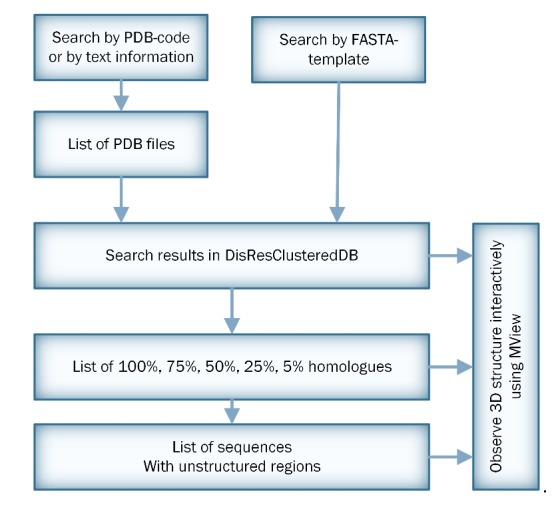
Scheme for server processing.

**Figure 4 molecules-25-01522-f004:**
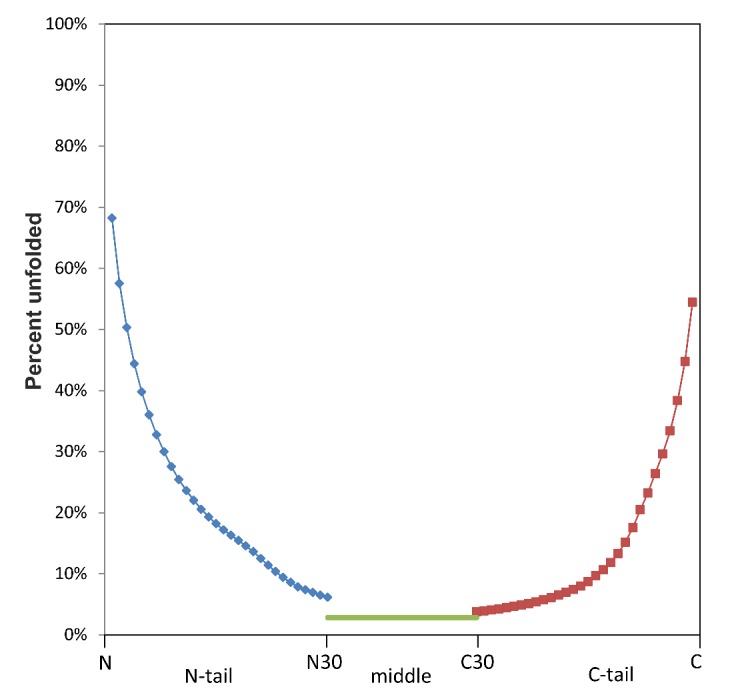
Statistics of all disordered residues from the clustered PDB at the 75% level of identity for the different positions of protein chains.

**Figure 5 molecules-25-01522-f005:**
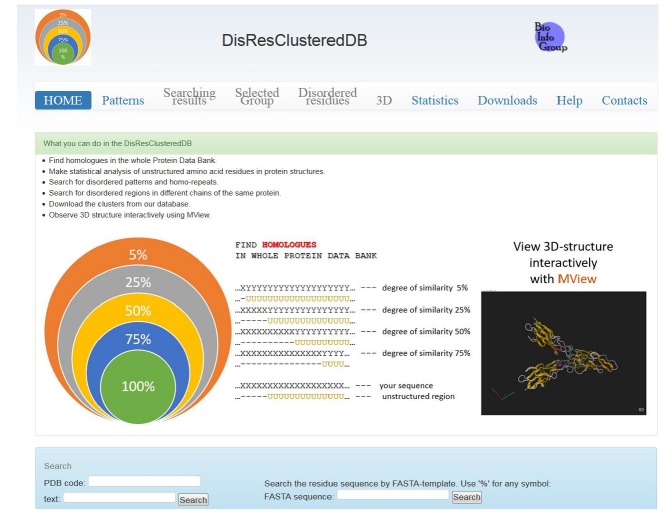
Common view of the database.

**Table 1 molecules-25-01522-t001:** Comparison of data for different AC2 patterns (pdb).

№	AC2 Pattern	N_u_	N_f_	N_u_–N_f_	N_C75_	N_prot_
Homo-repeat and adjacent amino acid						
1	00000001	342.6	166.6	176.0	73	101
2	01111111	349.3	175.8	173.6	79	113
	Sum	467.3	225.3	242.0	80	114
Internal repeat in the patterns						
3	00010001	499.1	17.3	481.8	56	66
4	01000100	384.9	46.4	338.6	61	75
5	00100010	355.3	47.7	307.6	56	65
6	01010101	369.7	66.3	303.4	53	80
7	01110111	336.1	46.3	289.8	51	62
8	01100110	35.4	12.6	22.8	7	10
9	00110011	32.4	19.6	12.8	7	11
	Sum	1040.0	228.2	811.9	147	196
Other patterns (part)						
10	00010000	1159.5	142.3	1017.2	125	163
11	00001000	1199.2	194.4	1004.8	131	171
12	00100001	982.8	81.4	901.3	84	111
13	01111011	865.0	63.7	801.3	87	110
14	01000010	765.8	53.7	712.2	87	114
15	00011111	663.7	105.1	558.6	122	163
16	00000111	257.9	33.5	224.4	48	60
17	00001111	236.2	38.0	198.2	42	52
18	00100100	307.6	167.7	139.9	50	78
19	01001001	261.5	132.3	129.3	38	49
20	01111100	127.6	8.2	119.5	22	24
21	00111101	182.9	65.1	117.8	32	37
22	01010100	132.9	22.7	110.2	23	27
23	00111110	129.6	24.0	105.6	24	27
24	01101101	228.4	124.8	103.6	35	54
	Sum	5014.3	2491.5	2522.8	857	1230
	Total sum	6033.1	2827.9	3205.2	971	1398

Notes: N_u_/N_f_ is the sum of the average number of unfolded/folded residues in the clusters with 75% identity (C75); N_C75_ is the number of the clusters with 75% identity (C75).

**Table 2 molecules-25-01522-t002:** Comparison of data for the different AC2 patterns.

№	Pair a.a.	N_u_	N_f_	N_u_–N_f_	N_C75_	Nprot
1	GlySer	2539.9	319.4	2220.6	248	312
2	GluAsp	258.9	42.5	216.4	32	59
3	GlyHis	162.7	9.2	153.6	28	31
4	SerHis	156.6	3.7	152.9	25	27
5	AlaPro	171.4	41.8	129.6	25	57
6	GlyLys	94.5	2.5	92.0	12	18
7	GlnPro	88.0	3.2	84.9	9	13
8	SerAsp	92.0	16.3	75.7	11	13
9	GluLys	86.0	11.0	75.1	11	14
10	GlyArg	79.1	14.4	64.6	8	12
11	AsnHis	58.6	0.0	58.6	6	11
12	AsnAsp	61.8	6.3	55.5	6	6
104	IleGlu	0.0	42.0	−42.0	5	7
105	LeuGlu	9.9	57.4	−47.5	9	12
106	AlaGlu	73.8	129.9	−56.1	25	35
107	IleAla	10.0	76.8	−66.8	12	15
108	ValAla	17.0	84.1	−67.1	14	22
109	AlaSer	53.7	135.3	−81.7	18	27
110	ValGly	7.3	91.0	−83.7	12	16
111	AlaArg	18.8	108.9	−90.1	15	21
112	LeuAla	94.9	226.0	−131.1	44	75
113	GlyPro	14.2	190.9	−176.7	16	21
	Total sum	6033.1	2827.9	3205.2	971	1398

**Table 3 molecules-25-01522-t003:** Comparison of data for the different versions of libraries of disordered patterns (L_min_ is the minimum length of pattern and L_max_ is the maximum length of pattern).

№	Year	Number of Unique Patterns	L_min_	L_max_
1	2010	109	6	24
2	2011	141	4	17
3	2012	171	4	21
4	2018	384	4	28
5	2019	518	4	90

**Table 4 molecules-25-01522-t004:** Connection between patterns and histidine tag.

Region	Number of Patterns
PC = 0% (all patterns are far from H4)	186
0% < PC < 33%	226
33% ≤ PC ≤ 67%	33
67% < PC < 100%	40
PC = 100% (all patterns are near with H4)	33

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
