# Peer review of "Disordered Residues and Patterns in the Protein Data Bank"

_molecules, 2020, doi:10.3390/molecules25071522_

Round 1

Reviewer 1 Report

The manuscript related to a new database for disordered residues is very interesting with a crucial topic and a medium level of presentation.

For my concern in the introduction should provide examples of the biological value to investigate rationally the role exterted by IDRs and IDPs with stuydy of Uversky of P. Tompa or recent review with focused on the challenging idea to pharmacologically target disordered protein fragments as Int J Mol Sci. 2015 Apr 2;16(4):7394-412. doi: 10.3390/ijms16047394.

Also to the conclusion a section related to the epxerimental application of this study should be added

Author Response

Answer: Thank you for the comments and suggestions. We have added 14 new references in the introduction and conclusion.

Reviewer 2 Report

The manuscript submitted by Lobanov and collaborators titled "Disordered Residues and Patterns in the Protein Data Bank" presents a new version of their library of disordered motifs. The manuscript is not clear and needs style and English editing. In the introduction, the authors must clarify similarities and differences between regions of low complexity and disordered regions. And what about the eukaryotic linear motif (elm)?.

In results and discussion, the authors said "The statistics was collected" (line 78) but it must be "The statistics were collected", for grammatic correctness, and "The statistics were calculated" for scientific correctness. Unfortunately, there are many of these that it is impossible to accept this manuscript for publication before extensive English editing.

Finally, similar to the introduction, the discussion needs to an extensive bibliographical revision about low complexity, elm and disordered regions in proteins  

Author Response

The manuscript submitted by Lobanov and collaborators titled "Disordered Residues and Patterns in the Protein Data Bank" presents a new version of their library of disordered motifs. The manuscript is not clear and needs style and English editing. In the introduction, the authors must clarify similarities and differences between regions of low complexity and disordered regions. And what about the eukaryotic linear motif (elm)?.

Answer:

Thank you for the comments and suggestions. In this paper, we introduced patterns consisting of two types of amino acids, which are motifs with low complexity. There were quite a few of them, 1214. In the results, we specially presented figure 2, which illustrates the connection of these patterns with disordered residues. It turned out that such motifs with a length of more than 7 residues are usually disordered. Therefore, we added such patterns in our library. We also tried to find functional significance for the most common motifs in the elm library. It is interesting to note that sequences with GlySer pattern are found both at the N- and C-ends of the protein chain, and in the middle without much preference. The bright example of GlySer patterns is FUS protein.

In results and discussion, the authors said "The statistics was collected" (line 78) but it must be "The statistics were collected", for grammatic correctness, and "The statistics were calculated" for scientific correctness. Unfortunately, there are many of these that it is impossible to accept this manuscript for publication before extensive English editing.

Answer: Thank you. We have corrected.

Finally, similar to the introduction, the discussion needs to an extensive bibliographical revision about low complexity, elm and disordered regions in proteins  

Answer: We have added 14 new references.